# Examining the Role of Anxiety and Depression in Dietary Choices among College Students

**DOI:** 10.3390/nu12072061

**Published:** 2020-07-11

**Authors:** Michelle M. Keck, Helize Vivier, Jeffrey E. Cassisi, Robert D. Dvorak, Michael E. Dunn, Sandra M. Neer, Emily J. Ross

**Affiliations:** 1Department of Psychology, College of Sciences, University of Central Florida, Orlando, FL 32816, USA; jeffrey.cassisi@ucf.edu (J.E.C.); robert.dvorak@ucf.edu (R.D.D.); michael.dunn@ucf.edu (M.E.D.); sandra.neer@ucf.edu (S.M.N.); ejayneross@gmail.com (E.J.R.); 2Independent Researcher; helizev@gmail.com

**Keywords:** college students, dietary intake, eating behaviors, mental health, sex differences

## Abstract

This study examines the role of anxiety and depression symptoms in predicting dietary choices in emerging adults while accounting for sex differences in these relationships. Participants were 225 English speaking undergraduates enrolled in a university in southeastern United States. Participants were recruited through an online research recruitment application utilized by the university. Participants volunteered for a two-phased anonymous survey monitoring the effects of eating habits and gastrointestinal health in young adults. As part of this effort, participants completed self-reporting measures related to anxiety and depression, as well as an automated, self-administered 24-h diet recall. Multigroup path analysis was used to test primary hypotheses. Overall, a decrease in total caloric intake and an increase in sugar consumption were found as self-reported symptoms of anxiety and depression increased. In addition, there were sex differences in the relationship between depression and food choices. Men consumed more saturated fat as well as less fruits and vegetables as self-reported symptoms of depression increased. Results suggest symptoms of depression are a greater risk factor for poor nutrition in male college students than females. The findings provide another justification to screen for psychological distress in student health services given the implications on behavioral lifestyle and health.

## 1. Introduction

### 1.1. College Students, Mental Health and Lifestyle

College students experience significant changes in their physical and social environment that lead to increased independence, autonomy, and responsibility [1,2]. These changes may contribute to the higher rates of anxiety and depression among college students compared to the general population [3]. A nationwide report by the CDC based on 2017 data [4] found that individuals between the ages of 18 and 25, the typical age range of college students, account for 12.4% of the suicides in the country. This age group also has the highest risk for serious suicidal intentions and attempts [5]. Results from the World Health Organization (WHO) world mental health surveys international college student project found that major depressive disorder (MDD) had the highest prevalence rate of mental health disorders (18.5%) among college students, followed by anxiety (16.7%) [6]. In addition, female gender was a risk factor for developing a mental health disorder among college students [6].

The increased rates of anxiety and depression among college-aged individuals are important given the risk that any psychosocial impairments that arise during this period may persist through adulthood [7,8] and are especially concerning given the evidence that enduring lifestyle habits are established during this period [2,9]. One lifestyle area that is of concern is dietary choices. Studies on college students show that they are at heightened risk for developing disordered eating and symptoms of an eating disorder [10], as well as substance use disorders [11]. The development of unhealthy dietary choices during this period can lead to weight gain, associated chronic diseases, and risky health behaviors such as substance misuse [2,9,12,13,14]. It is important, therefore, to identify factors that would affect dietary choices at this stage given the long-lasting effects and potential health risks associated with habits that are formed during this stage.

### 1.2. Anxiety, Depression and Diet

Studies show some support for a link between anxiety and diet, but the results are limited in both their scope and findings. One common finding is that there are sex differences in relation to diet and anxiety [15], though there is little consensus about the exact nature of this relationship. At the food group level, red meat consumption was found to have associations with anxiety symptoms in women, while legumes were associated with anxiety symptoms for men [16,17]. One study found that men were more likely than women to eat when anxious [18], while others reported contradictory findings [19,20]. The conflicting nature of the findings demonstrates a need to clarify the nature of this association.

The association between depression and diet, on the other hand, has been well-established [15]. Depression has been found to have associations with dietary patterns; specifically, healthier diets such as the Mediterranean diet were linked with lower depressive symptoms, while a Western diet is typically linked with greater risk for depression [21,22,23]. Studies that examined this association at the food group or nutrient level have found evidence that consumption of folate, oils, fish, soy, fruits and vegetables may be related to reduced depressive symptoms [15,22]. Processed foods, sugar, refined grain and high-fat foods, on the other hand, are associated with increased depressive symptoms [15,22]. In general, studies have found that individuals who are depressed are more likely to consume more calories, have higher Body Mass Index (BMI) and poorer diet quality [24]. Overall, there is strong evidence for a negative association between depression and healthy eating, as well as a positive association between depressive symptoms and caloric intake. Given the mixed evidence for both increased and decreased caloric intake when anxious or depressed [18,19,20,24,25,26], there is a need to clarify the exact nature of this relationship.

### 1.3. Biologic Sex and Diet

There are few studies examining the role of biologic sex in the relationship between anxiety, depression and dietary choices. This is an area that should be taken into consideration when conducting diet research due to the inherent differences in eating behaviors across sex. The National Health and Nutrition Examination Survey (NHANES) 2015–2016 report released by the USDA [27] showed that men across all ages consumed more calories and worse diet quality compared to women regardless of age. Specifically, women consumed less protein, sugar, and fat compared to men [28]. Women were also found to have denser nutritional consumption and better adherence to recommended dietary guidelines compared to men (i.e., higher heathy eating index (HEI) component scores [29]). Additionally, there is strong evidence for sex differences in food choices based on motivation [30,31,32,33,34,35,36]. In general, women tend to make food choices based on nutritional and health reasons while men are more likely to prioritize cost and taste [31,32,34]. In terms of food preference, women were more inclined toward sweet foods while men displayed a penchant for meat [30]. These motivational factors are substantiated by studies that have found higher consumption of sugar, fruits, and vegetables, as well as less salt and fat by women, compared to men [33,35,36]. This suggests that motivational factors that influence food choices account for the sex differences in diet quality. However, few studies examined the role of sex differences in emotions and dietary choices. As reviewed above, the findings have been contradictory [18,19,20]. Thus, there is much left to explore regarding the role of biologic sex in the influence of anxiety and depression on dietary quality and choices.

### 1.4. Gaps in the Literature

There are several challenges associated with summarizing and interpreting the literature on the effects of anxiety and depression on diet. One of these is the variety of assessment methods used across studies to assess diet [15]. This is problematic because each of these methods produces different metrics of diet [15]. There are also methodological differences regarding the level of diet analysis. Diet can be analyzed at the eating pattern, food group and nutrient levels, making cross-study comparison difficult [37].

Another challenge is that a thorough inventory of what is consumed by the individual is lacking in most studies [15,38]. Studies that use questionnaires that only examine select dietary components do not take into account the complexity of the human diet and assume that dietary components are independent of each other [15,38,39]. Twenty-four-hour recalls such as the automated self-administered 24-h recall (ASA24) provide complete dietary data that also allow for all levels of dietary analysis [39,40]. The ASA24 is a public web-based tool that provide researchers with nutrient and food group level information based on participants’ responses as well as information on social activity, food source and screen time related to eating [40]. Use of the ASA24 affords researchers a method to standardize data collection and food coding across different studies. Additionally, data derived from the ASA24 accounts for all foods consumed.

### 1.5. Study Purpose and Hypotheses

The purpose of this study was to investigate the relationship between anxiety and diet (i.e., dietary quality and choices), as well as depression and diet among college students. Additionally, the literature review indicated a need to account for the influence of biologic sex in this relationship. To test the following hypotheses, data were collected using the ASA24, which was then used to derive HEI total and component scores.

**Hypothesis** **1a.***There will be an inverse relationship between anxiety with diet quality, and total caloric intake*.

**Hypothesis** **1b.***There will be an inverse relationship between depression with diet quality and total caloric intake*.

**Hypothesis** **2a.***The interaction slope between anxiety with diet quality is expected to be steeper for men compared to women while the interaction slope between anxiety with total caloric intake is expected to be steeper for women*.

**Hypothesis** **2b.***The interaction slope between depression with diet quality is expected to be steeper for men compared to women while the interaction slope between depression with total caloric intake is expected to be steeper for women*.

**Hypothesis** **3a.***There will be an inverse relationship between anxiety with sugar and saturated fat*.

**Hypothesis** **3b.***There will be an inverse relationship between depression with sugar and saturated fat*.

**Hypothesis** **4a.***Slopes for the interaction between anxiety score with indices of diet quality are expected to be steeper for men compared to women*.

**Hypothesis** **4b.***Slopes for the interaction between depression score with indices of diet quality are expected to be steeper for men compared to women*.

## 2. Materials and Methods

### 2.1. Participants and Recruitment

Participants were English-speaking undergraduates between the ages of 18 and 25 who were enrolled in introductory psychology courses at a large university in the southeastern United States. Introductory psychology is a required course in the general education curriculum for most majors at this university. Therefore, the undergraduate population and all majors were well represented. Two hundred and sixty-one participants responded to the survey and were considered for this study.

### 2.2. Procedure

Participants were recruited via SONA, an online research recruitment application utilized by the university. The study was conducted in two phases. Phase I consisted of an online survey that collected demographic information as well as measures of anxiety and depression. Participants who completed Phase I were emailed an invite to complete Phase II. Phase II participants were provided with an external link to the ASA24 site and instructions for completing the food recall measure. The protocol (anonymous survey) received a waiver for the need for written informed consent, which was approved by the university’s Institutional Review Board (IRB; IRB#: SBE-18-14091).

### 2.3. Measures

The generalized anxiety disorder 7-item scale (GAD-7) was used as a measure of anxiety symptoms in this study. The GAD-7 is a widely used, brief, self-reporting measure that was developed to screen for anxiety symptoms in a medical or primary care setting and has been shown to have strong psychometrics [41,42]. Specifically, the GAD-7 has been found to show good internal consistency (α = 0.89) and construct validity in the general population [43]. It has also been validated for use in multiple studies [41,42,43]. Given the impact this study could have on primary care practices, the GAD-7 was determined to be the most suitable measure of anxiety for this study. The GAD-7 can be found in Appendix A.

The Patient Health Questionnaire 9-item scale (PHQ-9) was used as a measure of depressive symptoms in this study. The PHQ-9 is a widely used, brief, self-reporting measure that was developed to screen for depressive symptoms in a medical or primary care setting and has been shown to have strong psychometrics [44,45]. Specifically, the PHQ-9 has been found to have good internal validity (α = 0.89) and construct validity in the general population [44,45]. It has also been found to have good diagnostic properties across different medical care settings [46]. Given the impact this study could have on primary care practices, the PHQ-9 was determined to be the most suitable measure of anxiety for this study.

The automated self-administered 24-h recall (ASA24) was used to obtain raw dietary data for this study. The ASA-24 is administered online, and is a self-reporting 24-h food recall application developed by the National Cancer Institute (NCI) to measure daily nutritional intake [40]. The ASA24 is adapted from the United States Department of Agriculture (USDA) Automated Multiple-Pass Method (AMPM). The AMPM is a computerized and validated method for collecting interview-administered data. The primary distinction between the original AMPM and the ASA24 is the ASA24 has a dynamic guided interface that is respondent driven [47]. The ASA24 requires the participant to report meals, add details, and review and complete their responses. Probes are included for frequently forgotten foods and for an association with events in the past 24 h. It provides “food lookup tables” and the ability to search, add, change, or delete foods throughout the recall interview [48]. The process includes 5 steps. (1) a quick list collection of foods and beverages consumed the previous day, (2) a forgotten food probe, (3) time and eating occasion (e.g. breakfast, lunch, snack), and (4) a detailed cycle of the description of the food, amount, preparation, and additions to the food. Portion size is assessed using both pictures and measurement units. Finally, Step (5) is a probe inquiry asking the respondent for any additional entries. A final review prompts the respondent to review for validity or inconsistency related to their responses. The ASA24 data analysis files provide data on the following nutrients and food groups: macronutrients and energy, vitamins, minerals, carotenoids, fats and cholesterol, specific fatty acids, and other substances. Food categories include 4 fruit categories, 3 grain categories, 12 protein categories, 2 fat categories, 10 vegetable categories, 4 dairy categories, 1 added sugar category, and 1 alcohol category (number of drinks). The ASA24 automates food coding based on the food and nutrient database for dietary studies (FNDDS) [49]. The FNDDS provides researchers with detailed nutrient profiles and evaluation of food and beverages compared to FNDDS guidelines. The ASA24 has been evaluated against objective recovery biomarkers [50]. These findings indicated that the ASA24 provides a best estimate of dietary intake and has outperformed food frequency questionnaires. It is a feasible self-reporting measure for collecting dietary data when compared to objective biomarkers [50]. The ASA24 has been found to be comparable to an interviewer-administered 24-h recall regarding the accuracy of food reported in both adult and adolescent populations and has 80% accuracy in reporting rate when compared to actual food intake [51,52,53]. The accuracy of the ASA24 has similarly been found to be comparable to food journals [54]. The ASA24 provides multiple output tables for various levels of dietary analysis [55]. The Totals output was used to obtain HEI total and component scores, as well as total caloric intake. Information on the ASA24 output tables as well as details regarding the development and administration of the ASA24 can be found on the NCI website [55,56].

The healthy eating index (HEI) was used as a scoring index of diet quality in this study. It was initially developed to provide a guideline for professionals to use in assisting individuals with improving their diet quality [29]. It has since been used at the organization level to develop diet-related programs and policies for the general public and is the basis for federal nutrition education materials [29]. The HEI is a global index of healthy eating that quantifies a score by analyzing diet at the food group level [29]. The HEI score was used to calculate diet intake on a given day at person-level using data from the ASA24. Prior to calculating the HEI score, diet data must be mapped to food groups and listed by nutrients. The ASA24 disaggregates foods eaten into the food group components that are then used to calculate the HEI Component scores. The HEI consists of ten food group items, three of which are further broken down into sub-components (see Appendix A in Appendix A for breakdown of HEI components and scoring standards) [29,57]. HEI total and component scores for this study were derived from the ASA24 Totals output table using the HEI-2015–Per Day SAS macro developed by the National Institute of Health [58]. The macro also provided total caloric intake used to derive HEI total and component scores. The HEI defines diet quality based on nutrient density (i.e., nutrient content per 1,000 kcal) and is scored based on dietary recommendations from the U.S. Dietary Guidelines for Americans as recommended by the USDA [29]. These guidelines are required by the U.S. congress to be updated every five years, making the HEI an index that remains relevant and reflective of current evidence in nutritional research [59].

## 3. Results

### 3.1. Preliminary Analysis

Data from both phases were screened for missing data. Thirty-three participants were missing all relevant Phase I data (i.e., demographics, GAD-7 and PHQ-9 scores) and three participants were missing all Phase II data (i.e., ASA24). These participants were removed from the analysis, rendering a final sample size of two hundred and twenty-five.

Descriptive statistics were screened for skewness, kurtosis, and outliers (descriptive statistics are illustrated in Appendix A in Appendix A). Next, sex differences in race, marital status, college status, GAD-7 severity, and PHQ-9 severity were examined, as well as outcome variables. No sex differences were found regarding race, marital status and college status (illustrated in Table 1). Significant sex differences were found across GAD-7 and PHQ-9 severity levels, HEI total score, total caloric intake, sodium, and fruits intake (illustrated in Table 2). Correlation analysis indicated a significant correlation between the two predictor variables (*r* = 0.80, *p* < 0.05; illustrated in Appendix A in Appendix A). The strong correlation between GAD-7 and PHQ-9 scores are comparable to that found in other studies that used samples of healthy college students [60,61].

### 3.2. Primary Analysis

Multigroup path analysis (MGPA) using maximum likelihood estimation with robust standard errors to account for non-normal distributions in the primary variables [62] were conducted using MPlus 8.1. Path analysis provides testing of causal models by assessing the model fit with the data [63]. The inclusion of the multigroup option provides equivalence testing of different model pathways across biologic sex (Hypotheses 2a., 2b., 4a. and 4b.) [63]. The maximum likelihood estimation with robust standard errors was utilized in structural equation and path analysis to adjust for the use of continuous variables with non-normal distributions [64,65]. Model comparisons were determined using the loglikelihood ratio test (LRT; See Appendix B for formula) as recommended by Muthén and Muthén [66].

### 3.3. Primary Analysis: Anxiety and Global Diet Quality Indices Model

Both the freely estimated and fully constrained anxiety and global diet quality indices models were found to have good fit to the data. Examination of individual coefficients indicated a significant effect of GAD-7 score on total caloric intake but not HEI total score (summarized in Table 3). Specifically, there is a decrease of 30.16 calories with every 1-unit increase in GAD-7 score. These results support Hypothesis 1a regarding the effects of GAD-7 score on total caloric intake but not on HEI total score.

There were no significant differences between the between the freely estimated and fully constrained, Δ*χ*^2^ (2) = 0.16, *p* = 0.92, indicating no sex differences in the relationship (i.e., slopes) between GAD-7 score with HEI total score and total caloric intake. Thus, further examination between groups was not warranted. These results do not support Hypothesis 2a, as no significant sex differences in the slopes between GAD-7 score with HEI total score and total caloric intake were found.

### 3.4. Primary Analysis: Depression and Global Diet Quality Indices Model

Both the freely estimated and fully constrained depression and global diet quality indices model were found to have good fit to the data. Examination of individual coefficients indicated a significant effect of PHQ-9 score on total caloric intake but not HEI total score (summarized in Table 3). Specifically, there is a decrease of 27.44 calories with every 1-unit increase in PHQ-9 score. These results support Hypothesis 1b regarding the effects of PHQ-9 score on total caloric intake but not on HEI total score.

There were no significant differences between the freely estimated and fully constrained, Δ*χ*^2^ (1) = 0.35, *p* = 0.84, indicating that there are no sex differences in the association (i.e., slopes) between PHQ-9 score, HEI total score, and total caloric intake. Thus, further examination between groups was not warranted. These results do not support Hypothesis 2b, as no significant sex differences in the slopes between PHQ-9 score with HEI total score and total caloric intake were found.

### 3.5. Primary Analysis: Anxiety and Specific Diet Indices Model

Both the freely estimated and fully constrained anxiety and specific diet indices model were found to have good fit to the data. Examination of individual coefficients indicated a significant effect of GAD-7 score on HEI sugar component score but not HEI saturated fat component score. Specifically, there is a 0.16 decrease in HEI sugar component score for every 1-unit increase in GAD-7 score. These results support Hypothesis 3a regarding the effects of GAD-7 on HEI sugar component score but not for HEI saturated fat component score (summarized in Table 4).

There were no significant differences between the freely estimated and fully constrained models, Δ*χ*^2^ (10) =13.51, *p*= 0.20, indicating that there are no sex differences in the relationship (i.e., slopes) between GAD-7 score and HEI component scores. Thus, further examination between groups was not warranted. These results do not support Hypothesis 4a, as no significant sex differences in the relationship between GAD-7 score and HEI component scores were found.

### 3.6. Primary Analysis: Depression and Specific Diet Indices Model

Both the freely estimated and fully constrained depression and specific diet indices model were found to have good fit to the data. Examination of individual coefficients indicated a significant effect of PHQ-9 score on HEI sugar component score, but not HEI saturated fat component score. Specifically, there is a 0.17 decrease in HEI sugar component score for every 1-unit increase in PHQ-9 score. These results support Hypothesis 3b regarding the effects of PHQ-9 score on HEI sugar component score but not HEI saturated fat component score (summarized in Table 4).

There was a significant difference between the freely estimated and fully constrained models, Δ*χ*^2^ (10) =20.27, *p* = 0.03, indicating sex differences in the association (i.e., slopes) between PHQ-9 scores and HEI component scores. Thus, further examination between groups is warranted. These results show preliminary support for Hypothesis 4b in that there are sex differences in the relationship between PHQ-9 score and HEI component scores. Testing of each pathway revealed significant model differences when HEI component scores of saturated fat, fruits, and vegetables were constrained (summarized in Table 5). There were no significant model differences when HEI component scores for whole grain, dairy, fatty acid, sodium, refined grain, sugar, and protein were constrained (summarized in Table 5).

HEI component scores for whole grain, dairy, fatty acid, sodium, refined grain, sugar, and protein were constrained. The final model showed good model fit to the data, *χ*^2^ (7, *N* = 225) = 7.10, *p* = 0.42; CFI = 1.00; TLI = 1.00; RMSEA = 0.01 (90% CI = 0.00–0.12). Examination of the coefficients in the final model showed significant sex differences in the relationship between PHQ-9 score and HEI component scores of saturated fat, fruits, and vegetables. Specifically, the coefficients for HEI component scores of saturated fat, fruits, and vegetables were significant for men but not women (summarized in Table 6; see Appendix A in Appendix A for all final model coefficients). Additionally, the relationship between PHQ-9 score and HEI saturated fat component score was found to be direct for men and inverse for women (illustrated in Figure 1), while the relationship between PHQ-9 score and HEI component scores of fruits and vegetables were inverse across sex (illustrated in Figure 2 and Figure 3). There is support for Hypothesis 4b regarding the interaction between PHQ-9 score and HEI components of saturated fat, fruits and vegetables only. That is, the slopes for the interaction between PHQ-9 score with HEI component scores for saturated fat, fruits, and vegetables were steeper for men compared to women (summarized in Table 6), but not HEI component scores for whole grain, dairy, fatty acid, sodium, refined grain, sugar, and protein.

## 4. Discussion

### 4.1. The Present Study

There is strong evidence in the literature establishing associations between symptoms of anxiety and depression with dietary choices [1,15,20,26]. However, the magnitude of these associations has yet to be established [15,26]. One of the challenges associated with quantifying the effects of symptoms of anxiety and depression on diet are inconsistencies in the measurement of diet [15,38]. Additionally, insufficient attention has been paid to the role of biologic sex in the relationship between anxiety, depression, and diet. The current study addresses these gaps and contributes to our understanding by estimating the impact of anxiety and depression symptoms on dietary quality and choices among college students while accounting for sex differences in these relationships. Indeed, symptoms of anxiety and depression were associated with reduced total caloric intake and increased sugar consumption for all participants. Men, in particular, appeared to consume more fat, and fewer fruits and vegetables as symptoms of depression increased.

The first set of multigroup path analyses confirmed Hypotheses 1a and 1b regarding caloric intake only. That is, within the anxiety global diet quality indices and depression global diet quality indices models, GAD-7 and PHQ-9 scores had an inverse relationship with total caloric intake but not HEI total score. The analyses failed to confirm Hypotheses 2a and 2b in that no sex differences were found in the relationship between GAD-7 score with HEI total score or total caloric intake, nor in the relationship between PHQ-9 score with HEI total score or total caloric intake. Overall, the results of the multigroup path analyses with the anxiety and depression global diet quality indices models suggest that anxiety and depression decrease total caloric intake for all subjects. However, there are no significant differences in this relationship across biologic sex on these global measures.

The second set of multigroup path analyses confirmed Hypotheses 3a and 3b regarding sugar only. That is, for all subjects within the anxiety, depression, and specific diet models, GAD-7 and PHQ-9 scores had an inverse relationship with HEI sugar component score but not HEI saturated fat component score. The analysis also failed to confirm Hypothesis 4a in that no sex differences were found in the relationship between GAD-7 score and HEI component scores. Within the depression specific diet model, there were sex differences in the relationship between PHQ-9 score and HEI component scores of saturated fat, fruits, and vegetables. Specifically, greater declines in these food components were observed in men compared to women. Overall, the results of the multigroup path analysis with the anxiety and depression specific diet indices models suggest that while both anxiety and depression affect sugar consumption for both sexes, depression was associated with reduced consumption of saturated fat, fruits, and vegetables in men.

The findings that total calorie consumption and HEI sugar component score decreased with higher levels of anxiety and depression are particularly interesting when examined together. Lower HEI sugar component score indicates an increase in energy contribution of sugar in the total energy consumed. In other words, sugar accounted for more energy contributions with higher levels of anxiety and depression. One explanation for these findings is that sugar consumption increases at higher levels of anxiety and depression. Anxiety can induce physiological changes that curb appetite while depression can lead to a reduction in motivation for activity, including eating [26]. The availability and palatability of sweet foods make them an appealing and pleasurable option when individuals do not feel like eating but require energy [67]. Parental practices that encourage the use of sweet foods to regulate mood may also explain the increase in sugar consumption during times of psychological distress [68]. In short, while experiencing higher levels of anxiety and depression can cause a decrease in appetite, individuals may be more inclined to consume sweet foods when they do eat.

The results can also be attributed to a decrease in the consumption of low-sugar foods, especially considering the decrease in HEI component scores of saturated fat, fruits, and vegetables at higher levels of depression for men. A lower HEI saturated fat component score indicates an increase in the energy contribution of saturated fat in the total energy consumed while lower HEI component scores of fruits and vegetables indicate a decrease in fruits and vegetables consumption. In short, men who experienced higher levels of depression were more inclined to decrease total caloric intake as well as consumption of fruits and vegetables but not sugar and saturated fat. This is consistent with Grossniklaus, et al. [69]’s suggestion that sweet and fatty foods are more palatable, as well as Boek, Bianco-Simeral, Chan and Goto [31]’s finding that men are more motivated by taste when selecting foods.

Another interesting finding is that men with higher levels of depression were more likely to have worse diets (i.e., increased saturated fat, decreased fruit and vegetable intake) while women’s food choices were more stable. One reason for this sex discrepancy could be that women are more intrinsically motivated to select foods for health and nutrition while men’s food choices are driven by taste [31,32]. These motivations may endure with increased psychological distress, prompting women to continue to make healthier food choices while men select more palatable foods. Isasi, Ostrovsky and Wills [68] suggest that emotion regulation may play a role in the association between depression and dietary choices. Specifically, individuals with better emotion regulation abilities are more likely to have increased self-efficacy and subsequently, better diet quality. Individuals with poor emotion regulation, on the other hand, are more likely to experience depressive symptoms and decreased self-efficacy, leading to a lower likelihood of sustaining diet quality [68]. This is supported by an expectancy study done by Lyman [70], who found that individuals tend to prefer healthy food (e.g., fruits and vegetables) when they are experiencing positive moods and unhealthy foods (e.g., sugar and fat) when experiencing negative moods. Christensen and Brooks [71] suggest that consuming sweet and fatty foods trigger the release of opioids in the brain, improving mood and exponentially enhancing the palatability of the food. One possibility regarding the directionality of the relationship between mood and diet is that there is a cyclic effect of mood and food [68,70]. Specifically, while mood can affect food preferences, the foods consumed can also affect mood [68,70].

### 4.2. Clinical Impact

The findings regarding total calorie consumption are particularly important, given the implications for health. The decrease of 30.16 calories per 1-unit increase in GAD-7 score is equivalent to a decrease of up to 633.36 calories per day for someone with severe anxiety. The decrease of 27.44 calories per 1-unit increase in PHQ-9 score is equivalent to a decrease of up to 740.88 calories per day for someone with severe depression. This accounts for approximately 24%–41% of the recommended daily caloric intake for sedentary men and women (See Table 7 for comparison) and can have a severe impact on physical wellbeing. This is particularly concerning given the associated risks for malnutrition, metabolic disturbances, and issues related to fertility [72,73,74,75]. Malnutrition and its subsequent impact on health is particularly concerning given the findings regarding increased consumption of sugar and fat as well as decreased consumption of fruits and vegetables with higher levels of anxiety and depression.

The study also showed decreased HEI component score of sugar with increased levels of anxiety and depression as well as decreased component scores of saturated fat, fruits, and vegetables with depressed men. This decrease in HEI component scores indicates an increase in sugar and fat contribution to total energy consumed as well as a decrease in fruits and vegetables consumption. The comparison of consumption of these diet indices compared to the daily intake recommended by USDA guidelines can be seen in Appendix A in Appendix A.

The increase in sugar and saturated fat contribution to total energy consumed is concerning considering its impact on long-term health. Increased sugar consumption has been found to be associated with increased risk of cardiovascular disease, metabolic syndrome and obesity [76,77,78]. The significant decrease in fruit and vegetable consumption, on the other hand, has stronger implications on the reduction of diet-related diseases and mortality. Specifically, higher fruit and vegetable consumption has been theorized to be protective against cardiovascular diseases and certain types of cancers, including digestive cancer [79,80,81]. It has also been found to be associated with a lower risk of multimorbidity (i.e., having two or more chronic diseases), specifically those involving hypertension and diabetes [80].Though not a primary focus of this paper, we also found significant sex differences in self-reported symptoms of anxiety and depression. Specifically, women were more likely to report increased anxiety and depression compared to men. Interestingly, the higher rates of reported psychological distress among women did not affect their diet quality compared to men, who demonstrated poorer diet with increased depression. However, this sex discrepancy in reported distress and health-related issues has been well documented across a range of settings [82]. Thus, we expect the correlations found in our study between mental health reporting and dietary intake to be similar to that found in a clinical setting.

Overall, the results of this study indicate that depression may be more of a risk factor for poor nutrition compared to anxiety and that men’s nutrition is more affected by psychological distress. The results also suggest that caloric intake may be an area to address with individuals who identify more severe levels of anxious and depressive symptoms, given the negative health implications associated with it. The findings underscore the need to involve primary care (e.g., student health centers) in screening for psychological symptoms of distress given the association between increased risk of developing chronic diseases and poor nutrition [83]. This need is echoed in the recommendations made by the U.S. Preventative Services Task Force (USPSTF) to include screening for depression in all primary care practices [84]. Given the associations between depression and poor nutrition found in this study, there is a need for student health centers to adopt these screening methods, as primary care providers are more likely to detect and initiate early intervention, as patients suffering from psychological distress tend to present with somatic symptoms [85]. Within the student health network, referrals to dieticians can be made and nutritional guidelines developed to support preventative health measures during critical periods of development [59]. 

### 4.3. Strengths

Given the impact the present study has on primary care and health-related prevention, the screening measures that were used is considered a strength of the study, as they are developed for and commonly used in such settings. Additionally, the use of an evidence-based framework that is regularly updated (i.e., ASA24 and HEI) ensures that the body of evidence is relevant and effective to its time [86]. The use of the HEI in future studies and real-world settings is recommended as the dietary guidelines stipulated in the HEI is updated every five years as per the U.S. Congress [59]. The NCI also evaluates and makes updates to the ASA24 software to improve usability and efficacy [87]. The measures used in this study are also brief (i.e., GAD-7 and PHQ-9), user-friendly and cost-effective (i.e., ASA24), decreasing barriers and increasing interest in translating studies using such evaluations to real-world applications [86]. Finally, the analysis used provided evidence of directionality in the association between anxiety, depression, and diet that can be expanded upon and confirmed with longitudinal studies.

### 4.4. Limitations and Future Directions

Though this study contributes to the literature on the effects of anxiety and depression symptoms on dietary choices among emerging adults as well as sex differences in these relationships, several limitations exist that inform directions for future studies. While the rates of moderate and severe anxiety and depression in the sample collected here (16.8% and 19.5% respectively) were comparable to incidences reported by WHO for college students [16.7% and 18.5% respectively; 6], the absolute number of participants with moderate to severe anxiety and depression was too small to make confident inferences of population-level differences. A sample four to five times larger would yield more severely anxious and depressed participants and allow greater confidence in the estimates of impact on diet quality, total caloric consumption, and dietary choices. Additionally, dietary data was collected using a self-report measure. While the ASA24 accuracy is comparable to that of the gold standard interviewer-administered 24-h recall and food journals, self-reported data is always subject to recall bias. This could be addressed in future studies by using ecological momentary assessment (EMA).

Another limitation is the cross-sectional nature of the data. While the study provides insight on directionality, effects of anxiety and depression symptoms on diet as well as sex differences among these relationships, longitudinal assessment of diet is required to establish a dietary pattern and provide conclusive results [88]. Data was collected throughout the week and we did not account for the difference between weekday and weekend dietary intake in our analysis, due to sample size and power restrictions. This can be addressed in future studies using repeated measures that group dietary intake separately for weekday and weekend days. Additionally, our sample on average consisted of healthy weight adults (see Appendix A in Appendix A). This lack of variability in Body Mass Index (BMI) prevented us from accounting for the effects of weight in our analysis, a factor that has known associations with depression and caloric consumption [89,90]. However, the average BMI (*M* = 23.55) of our sample is consistent with the 2015–2019 reports released by the National College Health Assessment (NCHA; average BMI range 24.39–25.13) [91], indicating good representation of American college students’ BMI. Our overall findings regarding depression and caloric consumption in college students are pertinent to those in the average BMI range. This relationship may prove to be different for students with elevated BMI. Future studies should include related factors such as weight, income and physical activity using a longitudinal design to fully establish the effects of these factors on college students’ dietary intake [15,67].

Finally, the sample was recruited at a southeastern university in the United States, thus, the findings may not be representative of the culinary diversity across different areas of the country nor among individuals of different ethnicities. Further replication with larger samples across geographic regions is needed. Replication across cultural and culinary traditions would also provide insight on factors that would influence dietary choices when experiencing psychological distress.

## Figures and Tables

**Figure 1 nutrients-12-02061-f001:**
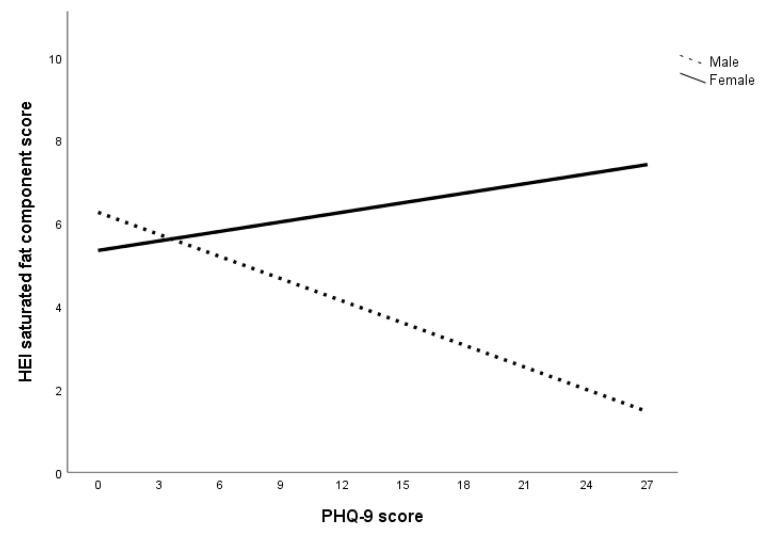
The relationship between depression and saturated fat consumption across sex. Note. Lower HEI saturated fat component score indicates higher consumption of saturated fat.

**Figure 2 nutrients-12-02061-f002:**
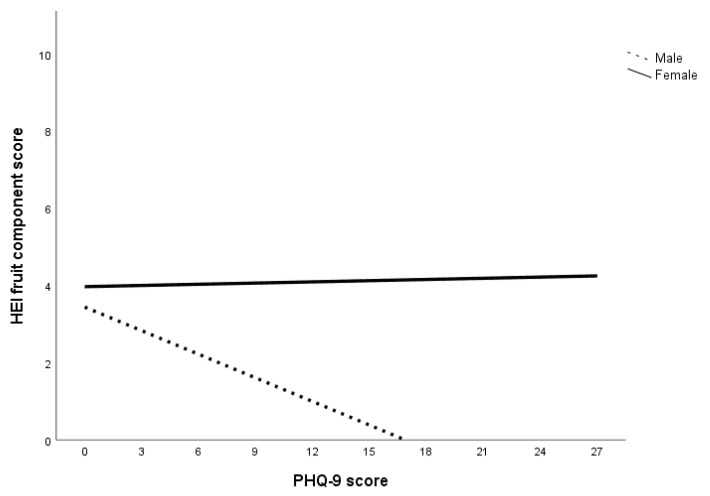
The relationship between depression and fruits consumption across sex. Note. Lower HEI fruits component score indicates lower consumption of fruits.

**Figure 3 nutrients-12-02061-f003:**
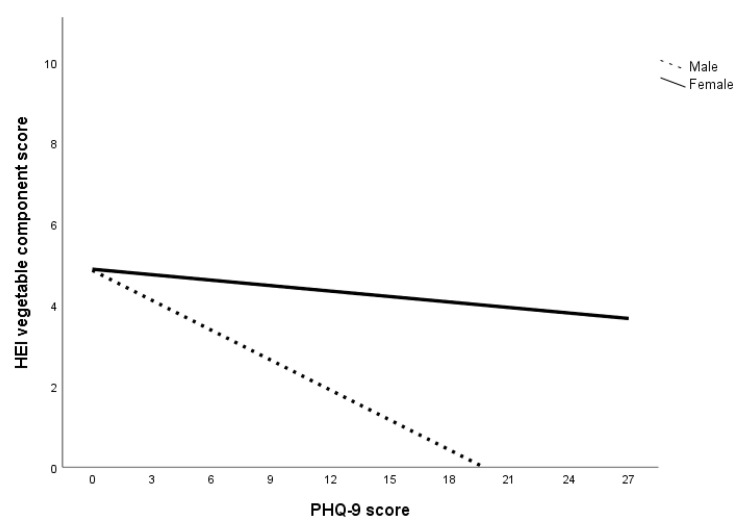
The relationship between depression and vegetable consumption across sex. Note. Lower HEI vegetable component score indicates lower consumption of vegetable.

**Table 1 nutrients-12-02061-t001:** Sex differences in participants’ race, marital status, college status, generalized anxiety disorder 7-item scale (GAD-7) severity and patient health questionnaire 9-item scale (PHQ-9) severity.

Variables	Male (*N* = 86)		Female (*N* = 139)			
	*N*	%	*N*	%	Chi-squared	*df*
**Race**					1.49	7
Native American	0	0.00	1	0.70		
Asian/PI ^a^	6	6.90	10	7.20		
Black	8	9.30	14	10.10		
White	49	56.90	75	53.90		
Mexican	3	3.50	3	2.20		
Puerto Rican	3	3.50	6	4.30		
Other Hispanic	9	10.50	18	12.90		
Mixed/Other	8	9.30	12	8.60		
**Marital Status**					0.03	1
Not married	85	98.80	137	98.60		
Married	1	1.10	2	1.40		
**College Status**					1.01	3
Freshman	52	60.50	84	60.40		
Sophomore	18	20.90	30	21.60		
Junior	12	13.90	15	10.80		
Senior	4	4.70	10	7.20		
**GAD-7 Severity**					11.61 ***	3
Minimal	57	66.30	68	48.90		
Mild	22	25.60	40	28.80		
Moderate	7	8.10	20	14.40		
Severe	0	0.00	11	7.90		
**PHQ-9 Severity**					10.03 *	4
Minimal	50	58.10	58	41.70		
Mild	27	31.30	46	33.10		
Moderate	6	4.30	22	15.80		
Moderately severe	3	2.20	8	5.80		
Severe	0	0.00	5	3.60		

^a^ PI: Pacific Islander * *p* < 0.05. *** *p* ≤ 0.001.

**Table 2 nutrients-12-02061-t002:** Sex differences in primary study variables.

Variables	Male	Female	
	Mean (SD)	Mean (SD)	F
Age	18.91 (1.42)	18.91 (1.50)	0.00
GAD-7 score	3.52 (3.82)	6.15 (5.52)	15.04 **
PHQ-9 score	4.58 (4.06)	6.68 (5.21)	10.19 ***
Whole grain ^a^	1.64 (2.75)	2.53 (3.61)	3.81
Dairy ^a^	5.18 (3.65)	5.19 (3.66)	0.00
Fatty acid ^a^	4.97 (3.77)	5.35 (3.98)	0.54
Sodium ^a^	2.72 (2.97)	4.16 (3.52)	9.99 **
Refined grain ^a^	4.71 (3.65)	5.09 (3.99)	0.51
Saturated fat ^a^	5.45 (3.72)	5.86 (3.54)	0.68
Sugar ^a^	6.95 (3.17)	7.29 (3.14)	0.62
Fruit ^a^	2.51 (3.75)	4.04 (4.26)	7.51 **
Vegetables ^a^	3.73 (3.41)	4.59 (3.49)	3.26
Protein ^a^	6.45 (2.91)	5.78 (3.20)	2.52
HEI ^b^ total score	44.29 (12.90)	49.86 (15.37)	7.85 **
Total caloric intake	2047.31 (875.61)	1691.34 (787.42)	9.96 **

^a^ HEI component scores. ^b^ HEI: Healthy Eating Index ** *p* ≤ 0.01. *** *p* ≤ 0.001.

**Table 3 nutrients-12-02061-t003:** Effects of anxiety on diet quality and total calorie consumption and depression on diet quality and total calorie consumption (Hypotheses 1a,b).

	b(SE)	*p*-Value
**GAD-7**		
HEI total score	0.41 (0.57)	0.48
Total caloric intake	−30.16 (10.67)	0.01 **
**PHQ-9**		
HEI total score	0.90 (0.83)	0.28
Total caloric intake	−27.44 (10.67)	0.01 **

** *p* ≤ 0.01.

**Table 4 nutrients-12-02061-t004:** Effects of anxiety on Heathy Eating Index (HEI) component scores for sugar and saturated fat and depression on HEI component scores for sugar and saturated fat (Hypotheses 3a,b).

HEI component scores	*b*(SE)	*p*-Value
**GAD-7**		
Sugar	−0.16 (0.05)	0.00 ***
Saturated fat	−0.01 (0.05)	0.91
**PHQ-9**		
Sugar	−0.17 (0.05)	0.00 ***
Saturated fat	0.03 (0.05)	0.65

*** *p* ≤ 0.001.

**Table 5 nutrients-12-02061-t005:** Model comparison between the freely estimated and individually constrained pathway models for step 4 of the depression and specific diet indices analysis.

HEI Component Scores	Δ*χ*^2^ (*df*)	*p*-Value
Whole grain	0.14 (1)	0.71
Dairy	1.38 (1)	0.24
Fatty acid	0.01 (1)	0.92
Sodium	2.03 (1)	0.15
Refined grain	0.12 (1)	0.73
Saturated fat	4.24 (1)	0.04 *
Sugar	0.00 (1)	1.00
Fruits	4.81 (1)	0.03 *
Vegetables	5.79 (1)	0.02 *
Protein	0.56 (1)	0.45

* *p* < 0.05.

**Table 6 nutrients-12-02061-t006:** Sex differences in the relationship between depression and food choices (Hypothesis 4b).

HEI Component Scores	*b*(SE)	*p*-Value
**Female**		
Saturated fat	0.07 (0.05)	0.19
Fruits	0.00 (0.07)	1.00
Vegetables	−0.04 (0.06)	0.53
**Men**		
Saturated fat	−0.15 (0.08)	0.04 *
Fruits	−0.19 (0.08)	0.01 **
Vegetables	−0.25 (0.07)	0.00 ***

* *p* < 0.05. ** *p* ≤ 0.01. *** *p* ≤ 0.001.

**Table 7 nutrients-12-02061-t007:** Comparison of USDA^a^ recommendation with the potential effects of severe anxiety and depression on caloric intake.

	Women	Men
USDA recommendation (calories)	1800–2000	2400–2600
Severe anxiety (−633.36 calories)	−31.67%–35.19%	−24.36%–26.39%
Severe depression (−740.88 calories)	−37.04%–41.16%	−28.50%–30.87%

^a^ USDA: United States Department of Agriculture

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
