# Peer review of "Examining the Role of Anxiety and Depression in Dietary Choices among College Students"

_nutrients, 2020, doi:10.3390/nu12072061_

Round 1

Reviewer 1 Report

Summary: Dietary choices can be influenced by mental health. College-aged young adults may be a unique population that is experiencing heightened anxiety and depression, but also responsible for making independent food choices. This is an important time period to study these relationships. The purpose of this study was to examine the relationships among depression, anxiety, sex, and diet quality in college students. The authors used the ASA-24 to collect detailed dietary intake data and then use the Healthy Eating Index guidelines to examine consumption in different food categories and overall diet quality. They found that there were some relationships between HEI component scores and depression/anxiety and that these relationships tended to be observed in males. Females were more likely to have stable food/energy intake across a spectrum of scores. These findings are interesting, but there is no consideration of BMI or weight status in this study and limited information on the way in which the HEI values are determined.

  • Overall, there is a general lack of review of the relationship between mental health and weight status, which is a robust literature. It is difficult to tease apart the impact of the reciprocal relationship between mental health and weight from the impact of diet quality and intake on weight.  
  • The hypothesis statements use the names of surveys, but these instruments have not been introduced at this point, so the hypotheses are not clear to readers who are unfamiliar with these questionnaires. It would be better to provide more detail (e.g. “There will be an inverse relationship between depression, as assessed by PHQ-9, and HEI score”).
  • The descriptions of the questionnaires used is brief and tend to end with the statement about the psychometrics before “strong”. This is a subjective term. It would be better to present the validity and reliability data for each as well as an explanation for why these instruments were chosen. This is especially important given that one of the weaknesses that was identified is the use of different measures across studies.
  • There is no information provided about the weight status of participants. Given the relationship between weight status and depression, it is important to understand the extent to which these relationships are explained by difference in BMI or whether there are interactions between sex and weight status. Although self-reported height and weight have limitations, these data are critical to the interpretation and clear understanding of the relationships outlined here.
  • Self-reported dietary intake data, even using systems like the ASA24 that improve accuracy using automated probing questions about the details of each food, are subject to a large amount of error in reporting. The authors should examine these data for underreporting and determine what percentage of the dietary data can be considered accurate.
  • One of the gaps identified here is that other studies have used dietary analyses that have focused on discrete dietary components as opposed to examining complete dietary intake data. However, this study took the results from a complete dietary recall and condensed it to discrete categories. It is unclear how this is an improvement, in particular because no data (such as macro and micronutrient intake) were used directly from the ASA-24.
  • Dietary recalls are typically conducted on multiple days and include at least one weekend day to account for differences in intake on weekdays versus weekends. There is limited information provided on the protocol used here, but it reads as though participants only completed a single dietary recall. It also does not appear that there have been efforts to take into account whether this recall was taken on a weekday or a weekend.   More information should be provided about the way in which dietary data were collected and adjustments made to the analysis to take into account whether the recall is a weekday or weekend day.
  • The sex differences here are interesting because they extend beyond typical sex difference in energy and macronutrient intake and describe the relationships between these and mental health. Given that there were sex differences in the scores on these questionnaires, how much of these relationships can be explained by women being more likely to accurately report mental health issues compared with men or that men on the higher end of the scale may be outliers?
  • More detail should be provided for how the foods were coded using the HEI. For example, how are combination foods (e.g. pizza, lasagna, salad?) dealt with?   How are snack foods classified?   Overall, I am having difficulty understanding how the ASA-24 data were used to determine the HEI component scores.

Minor Comments:

  • Please remove the “How to use this template” instructions.
  • Data was collected (line 119) should be changed to “data were collected”.
  • There is an asterisk out of place in Table 1

Author Response

Dear Editor and Reviewers,

We have endeavored to make the changes and edits you suggested. Thank you, we believe the paper is a much better product now given your feedback. We did not include obesity in our analysis for several reasons. To do that is beyond the scope of our study’s focus and our sample size is too small to conduct a recursive analysis. Also, our careful examination of BMI suggests that our dataset may not be optimal to analyze obesity. We do acknowledge that is a potential area for future work. Additionally, we expanded on the information regarding our measures. The ASA24 and HEI are measures that were developed by the National Cancer Institute and National Institute of Health based on guidelines set by the U.S. Department of Agriculture. To fully expand on the details regarding food coding and data conversion would be beyond the scope of our study. Thus, we have provided links to the ASA24 and HEI website whenever possible. Please see the attachment for full details. Thank you again for your contributions to this project.

Authors

Reviewer 2 Report

The purpose of this study was to investigate the relationship between anxiety and diet, as well as depression and diet among university students and the influence of biological sex on this relationship. To do this, food intake and health score data was collected using the ASA24, which was then used to obtain total scores and components of HEI, two well-considered methods.
Methodological errors that can be modified:
BMI data should have been collected, obese people consume more calories, have a higher body mass index (BMI) and poorer diet quality. There is strong evidence of a negative association between obesity and healthy eating, as well as a positive association between obesity and calorie intake. In addition, it is also important to know the income, since the economic level is a relevant factor for the choice of food. "People tend to choose foods based on nutritional and health reasons, cost, taste, ... not only for reasons of anxiety or depression"
The results regarding total calorie consumption should be adjusted according to daily physical activity. The daily caloric intake for the sedentary is much less than for an athlete.

The 7-item scale of generalized anxiety disorder (GAD-7) and the 9-item scale of the patient's health questionnaire (PHQ-9) can be seen in supplementary materials.
Depression is said to have associations with dietary patterns; Healthier diets, such as the Mediterranean diet, could be measured with the Mediterranean Diet Quality Index (KIDMED), a simple to use questionnaire that could complete your study.

Author Response

Dear Editor and Reviewers,

We have endeavored to make the changes and edits you suggested. Thank you, we believe the paper is a much better product now given your feedback. We did not include obesity, income and physical activity in our analysis for several reasons. To do that is beyond the scope of our study’s focus, and our sample size is too small to conduct a recursive analysis. Also, our careful examination of BMI and income suggests that our dataset may not be optimal to analyze obesity. We do acknowledge that these are potential areas for future work. Please see the attachment for more details. Thank you again for your contributions to this project.

Authors

Round 2

Reviewer 1 Report

I thank the authors for their detailed responses to my prior critiques.  While I realize that much of what I asked to be considered is beyond the scope of the study.  My largest concern had to do with the fact that BMI of the participants was not considered in the analysis.  While I understand that including BMI would reduce power, when a paper is exploring energy and nutrient intake and relationships with mental health, most readers will ask about weight status.

My remaining concern has to do with how BMI is handled even in the revised manuscript.  There is no mention of weight or BMI except one sentence at the end about how future studies should examine BMI.  I urge the authors to report the average BMI of the participants in the study (in Table 1 and/or Table 2), to include lack of consideration of BMI as a limitation and explain that because the hypotheses of the study were not related to BMI and because the sample is, on average, within the healthy weight range, there may not be enough variability in BMI to find any differences.  In sum, a more robust discussion of BMI in the paper is needed.  
